# Polymer microbubbles as universal platform to accelerate polymer mechanochemistry

Jilin Fan[1,2], Regina Lennarz [3], Kuan Zhang [1,2,4], Ahmed Mourran [2], Jan Meisner [3], Mingjun Xuan [1,2,4] ✉, Robert Göstl [1,2,5] ✉ & Andreas Herrmann [1,2] ✉

The flow-induced activation of mechanophores embedded in linear polymers by ultrasound (US) suffers from slow mechanochemical conversions at the commonly used frequency of 20 kHz and in many cases remains ineffective with higher MHz frequencies. Here, we present polymeric microbubbles (PMBs) as a platform that accelerates the mechanochemical activation of several mechanophores under both 20 kHz and MHz irradiation. MHz irradiation generated by biocompatible high-intensity focused US (HIFU). Through their pressure-sensitive gas core, PMBs act as acousto-mechanical transducers for the transformation of sound energy into stretching and compression forces as well as fracturing the polymer shell by the volume oscillation of PMB. We investigate three different mechanophores among which one flex-activation derivative was unexpectedly activated by US. Through a combination of experiments and computation, we find that PMBs likely exert compressive force onto the copolymerized mechanophores rather than the typical elongational forces solvated chain fragments experience in flow. We thereby underscore the mechanochemical properties of the PMB platform and its versatility for accelerated mechanochemical transformations with a perspective on biomedical applications.

Stimuli-responsive polymers that elicit a desired function when subjected to a specific chemical or physical stimulus are a subject of great interest[1–5]. Particularly, chemical transformations induced by mechanical force have received attention, since the force-activated alteration of polymer structures[6] or the generation of chemical functions for catalysis[7–10], sensing[11–13], cargo delivery[14–17], or soft robotics[18] promise exciting future applications. In this light, US has become an accessible and widely used tool in polymer mechanochemistry due to its non-invasive nature, spatiotemporal control, and ease of use[19–21] for applications in mechanochemical synthesis[22] and activation of material functions[15,23–25]. Because linear mechanophore-centered polymer chains conduct mechanical force along their backbone[26] and are straightforward to analyze by solution-borne methods, they are commonly employed to benchmark mechanophores[27–31]. Recent research shows that the architecture of the mechanophore-bearing polymer is an important parameter that influences the mechanochemical efficiency. Therefore, helical polymers[32], cyclic polymers[33,34], star polymers[35,36], polymer bottlebrushes[37–39], dendronized polymers[40–42], and polymer microgels[43–45] have been investigated.

Another architecture are polymer-shelled gas bubbles, which exhibit special topological characteristics, such as the absence of solvated end groups and restricted conformational freedom. PMBs are an established modality in theranostic biomedicine and have been used to enhance the US response for imaging, drug delivery, and sensing[46–49]. However,

[1]Institute of Technical and Macromolecular Chemistry, RWTH Aachen University, 52074 Aachen, Germany. [2]DWI – Leibniz Institute for Interactive Materials, 52056 Aachen, Germany. [3]Institute for Physical Chemistry, Heinrich Heine University Düsseldorf, 40225 Düsseldorf, Germany. [4]Wenzhou Institute, University of Chinese Academy of Sciences, 325001 Wenzhou, China. [5]Department of Chemistry and Biology, University of Wuppertal, 42119 Wuppertal, Germany. ✉e-mail: xuan@wiucas.ac.cn; goestl@uni-wuppertal.de; herrmann@dwi.rwth-aachen.de

mechanochemical transformations in PMBs have remained unknown until we recently reported the US-induced cleavage of disulfide mechanophores[50]. There, PMBs have served as acousto-mechanical transducers during the sonication, and subsequent disruption of the bubble shell was accompanied by mechanophore activation. However, the generality of this system remained unclear and we were unable to utilize biocompatible MHz US. Both aspects are crucial for the successful application of the PMB concept in a biomedical context.

Here, we show that the microfluidically engineered PMB system can accelerate the activation of several exemplarily chosen mechanochemical motifs for molecular release. Unexpectedly, we find that even the flex activation, which was previously impossible to achieve by ultrasonication, is enabled using PMBs. In addition, we investigate biocompatible MHz HIFU for the activation of PMB cargoes.

Therefore, uniformly sized, $N_2$-filled PMBs are prepared by microfluidics (Fig. 1a). Mechanophores are copolymerized into the PMB shell and activated with 20 kHz US and MHz HIFU (Fig. 1b). We demonstrate mechanochemical transformations in PMBs by four strategies (Fig. 1c). Firstly, disulfide mechanophores and subsequent molecular release are achieved by thiol-initiated Michael addition and retro Diels-Alder (rDA) reaction consolidating our previous `findings. Secondly, the fluorophore umbelliferone (UMB) and the active pharmaceutical ingredient camptothecin (CPT) are released by thiol/disulfide exchange and intramolecular cyclization, which also bases on the activation of disulfide mechanophores[15,51,52]. Third, we release a fluorogenic molecule from a masked furfuryl carbonate by mechanochemical rDA reaction and an autocatalytic effect, as principally conceived by Robb and coworkers[53]. Lastly, we achieve US-induced flex activation of a furan Diels-Alder adduct, which was pioneered by Boydston and coworkers[54–58]. These strategies underline the general applicability of the PMB system and even show that otherwise unobtainable mechanochemical reactions can be performed using US.

## Results

### Disulfide mechanophore activation for dansyl-fluorophore release

Previously we demonstrated the 20 kHz US-triggered disulfide mechanophore activation in microbubbles[50]. Hence, we first used this established system to understand the effect of HIFU in the MHz regime on the PMBs. The microfluidically engineered PMBs are composed of a polymer network shell consisting of poly(propylene glycol) diacrylate (PPGDA), mechanoresponsive crosslinker and probes or cargo molecules. A thiol-sensitive probe quantifying mechanochemical disulfide scission by fluorescence, a masked dansyl-fluorophore, was copolymerized into the polymer shell through photo-crosslinking. In addition, bis(2-methacryloyl)oxyethyl disulfide (N1-CL) was used as the mechanoresponsive crosslinker. US-induced deformation and fracture of the PMBs would expectedly lead to disulfide activation and the free thiols would react with neighboring masked dansyl-fluorophore molecules releasing the dansyl-fluorophore (N1-A3) via rDA reaction.

The gram-scale syntheses of the probe N1-A3, its complementary dienophile N1-B3, and the resulting Diels-Alder adduct monomer N1-DA are detailed in Supplementary Fig. 1[59]. Before copolymerization into the PMB shells, the reactivity of the constitutional isomers N1-DA1 and N1-DA2 toward thiols was verified by mixing with 2-mercaptoethanol for 4 d and subsequent fluorescence spectroscopy (Supplementary Fig. 8). Due to a higher reactivity, N1-DA2 was used as the masked dansyl-fluorophore (MDF) to prepare PMB-MDF.

Then, PMBs were prepared in a microfluidic device by gas-in-oil-in-water (G/O/W) double emulsion (Fig. 2a and Supplementary Movie 1). A UV lamp was set up surrounding the outlet tube for the PMB shell polymerization, which stabilized the gas ($N_2$) core. The shape of the PMBs was characterized by scanning electron microscopy

(SEM) and optical microscopy (Fig. 2b, Supplementary Fig. 9) with an average PMB size of ~29 μm (Fig. 2c). PMB samples were then treated with 20 kHz, 0.68 MHz, 1.52 MHz, and 2.6 MHz US (Fig. 2d, Supplementary fig. 11). At 20 kHz and an intensity of $I = 12\,W\,cm^{-2}$, sonication for 5 min led to fracture of the PMBs and the fragments sunk to the bottom of the tube. Under HIFU irradiation, PMB destruction commenced at a power intensity of $75.5\,W\,cm^{-2}$ for 0.68 MHz, $134.2\,W\,cm^{-2}$ for 1.52 MHz irradiation, and $1208\,W\,cm^{-2}$ for 2.6 MHz irradiation (Supplementary fig. 11). Microscopy showed that the PMB fragment size gradually decreased with sonication time at 20 kHz while comparably larger fragments remained with HIFU (Fig. 2e).

To quantify the mechanophore activation and fluorophore release (Fig. 2f), the suspensions of PMB fragments after 15 min sonication runs were collected, filtered, and fluorescence was measured (Fig. 2g). Approximately 23% of the copolymerized fluorophores were released at 20 kHz and 12 W cm⁻². When $I$ was decreased to 1.0 W cm⁻², we found that the release accordingly decreased to around 10% (Fig. 2h). Notably, more than 10% release was achieved when HIFU at 0.68 MHz and 134.2 W cm⁻² was used while no significant release was found below 33.6 W cm⁻² (Fig. 2i). Fluorophore release dropped to 5% when HIFU at 1.5 MHz and 134.2 W cm⁻² was used (Fig. 2j) and to <1% release for HIFU at 2.6 MHz and the same acoustic intensity likely due the absence of PMB fracture and fragmentation (Fig. 2k).

To investigate if the PMB deformation without fracture led to mechanophore activation, 5 min sonication at 20 kHz with a very low $I$ of 0.2 W cm⁻² was performed. Although under these conditions no bubble fracture was observed, and we consequently inferred that PMBs were only deformed, we found around 3% disulfide mechanophore activation (Supplementary fig. 18). In combination with the results obtained in Fig. 2e, h, indicating that the subsequent sonication of PMB fragments also led to mechanophore activation after fracture, we infer that mechanophore activation occurs in all three possible mechanochemical scenarios: PMB deformation, fracture, and subsequent secondary fragmentation.

As negative control for non-specific activation, we prepared PMBs without disulfide crosslinker (Supplementary fig. 19) and expectedly did not observe fluorescence after sonication (Fig. 2l). Furthermore, solid-core microgels were prepared by microfluidic single emulsion (Supplementary Fig. 20 and Supplementary Movie 2), serving as negative control regarding the role of the gas core of the PMBs. Supplementary Fig. 21 shows the optical micrograph of these microgels with an average diameter of ~ 26 μm. Upon sonication conditions identical to those of the PMBs, only a very weak increase of fluorescence was observed (Supplementary Fig. 22) corresponding to <0.1% release (Supplementary Fig. 23). The absence of morphological changes in the microgels after sonication indicated the important role of the gas core for mechanochemical activation (Supplementary fig. 23).

### Disulfide mechanophore activation for UMB and CPT release

In addition to the Michael addition of thiols with subsequent rDA release, thiol/disulfide exchange was activated mechanochemically in PMBs. Therefore, UMB (fluorophore) and CPT (drug) were selected as the functional target molecules[51]. The respectively carbonylated monomers N2-A3 and N2-B3 (Supplementary Fig. 2) would undergo intramolecular 5-exo-trig cyclization after mechanochemically initiated thiol/disulfide exchange releasing both UMB and CPT.

First, PMB-UMB was prepared from N2-A3 (Fig. 3a, b) and US treatment qualitatively confirmed UMB release by fluorescence turn-on (Fig. 3c, d). The maximum UMB loading was established by reduction of the disulfides and subsequent fluorescence spectroscopy (Supplementary Fig. 24). 15 min sonication runs of PMB-UMB at 20 kHz and 12 W cm⁻² released 19% UMB (Fig. 3e) while 0.68 MHz and 134.2 W cm⁻² led to 8% UMB release (Fig. 3f). Proton nuclear magnetic resonance ($^1H$ NMR) spectroscopy and mass spectrometry (MS)

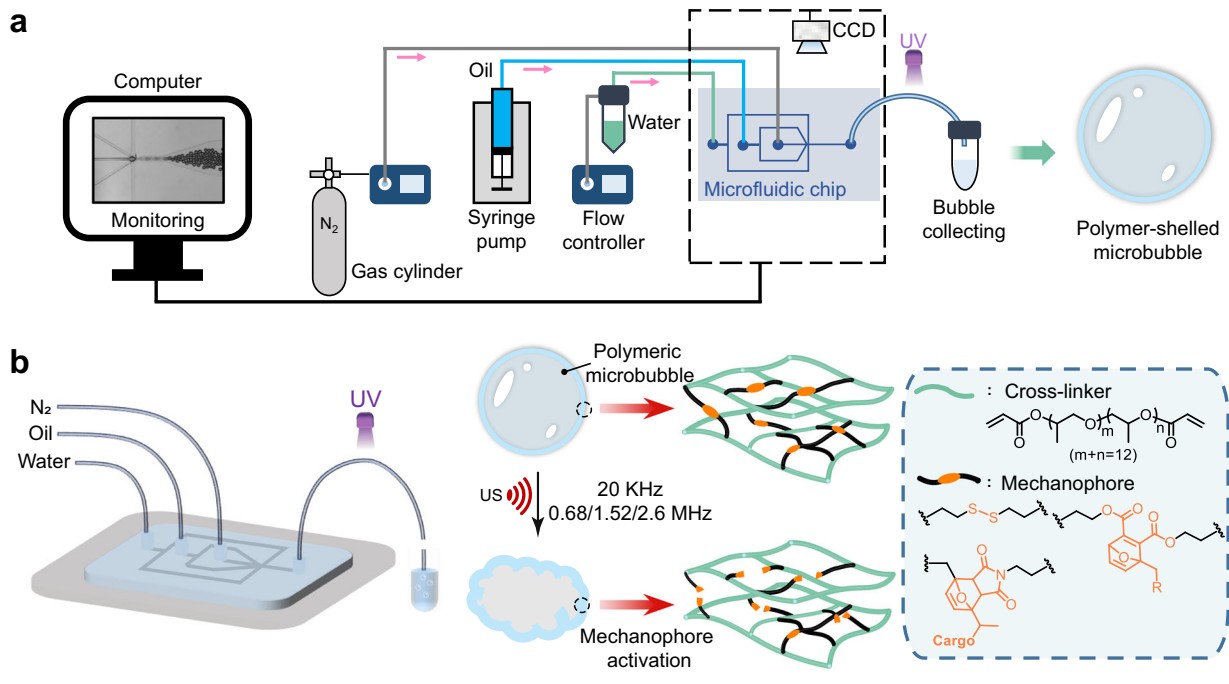

**Fig. 1 | Schematic illustration of the various mechanophore activation processes possible in PMBs.** **a** PMB fabrication using microfluidics including the employed setup and chip design. UV: UV irradiation, CCD: Charge-coupled device camera. **b** The preparation and mechanochemical activation of PMBs upon exposure to US. **c** The illustration of US-induced activation of diverse mechanophores in PMBs.

measurements of sonicated PMB-UMB cross-validated the successful release (Fig. 3g, Supplementary Fig. 27).

Having established the molecular proof-of-concept with UMB, pharmacologically active PMB-CPT was synthesized from N2-B3 (Fig. 3h) and its activity after sonication was investigated by MTS proliferation assays with HeLa cells (Fig. 3i). Since CPT is also fluorescent, this mode of detection was used to qualitatively confirm CPT release in conjunction with ultra-high performance

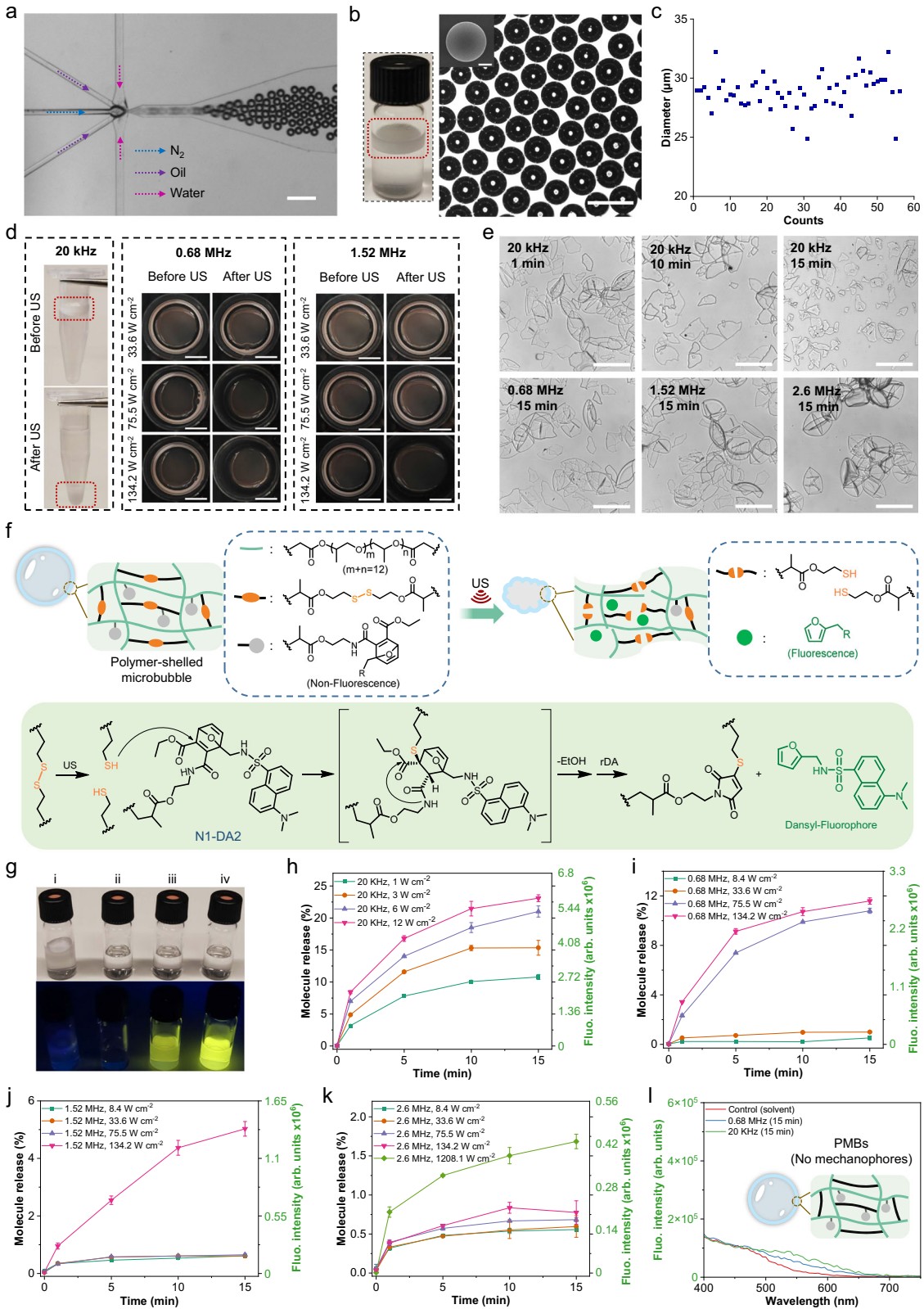

liquid chromatography-mass spectrometry (UPLC-MS, Fig. 3j). After 15 min sonication, 18% CPT were released at 20 kHz and 12 W cm$^{-2}$ (Fig. 3k) while 8% at 0.68 MHz could be achieved. Incubation of the sonicated PMB-CPT solutions with HeLa cells showed considerable lower IC$_{50}$ of PMBs after sonication than before while highlighting the biocompatibility of the PMB system and of the mechanophore-free PMB fragments (Fig. 3l).

## 2-Furylcarbinol Diels-Alder adduct mechanophore activation for pyrene fluorophore release

For the introduction of 2-furylcarbinol Diels-Alder adduct mechanophores, pioneered by Robb and coworkers[53,60], into PMBs, we designed and synthesized the masked furfuryl carbonate (**N3-A5**) containing a fluorogenic pyrenebutanol (PBL) payload (Fig. 4a, Supplementary Fig. 3). Subsequent to the mechanochemical scission of

**Fig. 2 | The activation of disulfide mechanophores and dansyl fluorophore release via rDA from PMB-MDF. a** The production of PMBs in microfluidic chip, scale bar: 100 μm. The experiment was repeated three times, and similar images were obtained. **b** The collection of PMBs in a glass vial and the associated optical micrograph (scale bar: 50 μm). Inset: SEM image, scale bar: 10 μm. The experiment was repeated three times, and similar images were obtained. **c** Diameter distribution of the PMBs corresponding to Supplementary fig. 9. **d** The response of PMBs to 20 kHz (12 W cm$^{-2}$), 0.68 MHz, or 1.52 MHz US for 5 min, scale bar: 5 mm. **e** Fragments of PMBs, with 20 kHz US (12 W cm$^{-2}$) or HIFU (134.2 W cm$^{-2}$ for 0.68 and 1.52 MHz, 1208 W cm$^{-2}$ for 2.6 MHz), scale bar: 50 μm. The experiment was

repeated three times, and similar images were obtained. **f** Scheme of mechanophore activation and fluorophore release. **g**, Photographs of samples under daylight and UV light ($\lambda_{exc}$ = 365 nm), (i) PMBs without sonication. (ii) MeCN/H$_2$O blank. (iii) The filtered solution of sonicated PMBs (0.68 MHz, 15 min). (iv) The filtered solution of sonicated PMBs (20 kHz, 15 min). **h–k** Fluorophore release from PMBs under 20 kHz, 0.68 MHz, 1.52 MHz, and 2.6 MHz irradiations. Data are presented as mean values +/− the standard deviation. $N$ = 3 independent sonications. **l** Fluorescence spectra of sonicated PMB control without disulfide. Data are presented as mean values +/− the standard deviation.

the furan-maleimide Diels-Alder adduct, PBL would be released based on the instability of furfuryl carbonate in polar protic solvents.

First, PMBs were prepared from **N3-A5** (PMBs-PBL) and analyzed by optical microscopy (Fig. 4b). The latent instability of the small molecule **N3-A4**, and hence its capability for molecular release, (Fig. 4c) were verified in a mixture of MeCN-d$_3$, MeOH, and H$_2$O (3:1:0.5, v/v/v) while monitored by $^1$H NMR spectroscopy. The decomposition process of **N3-A4** was complete within in 4 d at 37 °C accompanied by the formation of **N3-A3** and PBL (Fig. 4d) and negligibly slower at 23 °C (Fig. 4e and Supplementary fig. 31).

The fluorogenic properties of PBL allowed the observation of the mechanochemical release process by fluorescence spectroscopy (Fig. 4f, g). Around 24% PBL were released from PMBs-PBL using 20 kHz US at 12 W cm$^{-2}$ for 15 min (Fig. 4h). The PBL release expectedly correlated positively with both sonication time and US intensity. More than 10% released PBL were obtained using 20 kHz US at 1.0 W cm$^{-2}$ while 9% could be measured using 0.68 MHz at 134.2 W cm$^{-2}$ (Fig. 4i).

## Flex-activation of mechanophores to release dansyl-fluorophore

The overwhelming majority of mechanophores are activated by the direct scission of bonds force-coupled along the pulling vector of the polymer chain. One notable exception are flex-activated mechanophores where the scission of bonds and the subsequent release of small molecules are induced by a distortion of adjacent bond angles[55,57,58]. However, up to today these could only be activated in bulk polymers by compression with single digit percentage released fractions[55,57,58] where double network hydrogels are a notable exception releasing ~ 20% of their payload[61]. Conversely, the US-induced flex-activation has not been reported.

To achieve flex-activation in PMBs, we synthesized the required oxanorbornadiene-based mechanophore **N4-A3** incorporating furylated dansyl as fluorogenic probe cargo (Supplementary Fig. 4). This was then either appended with initiator moieties for controlled radical polymerization yielding **N4-C4** for linear control chains or with acrylates for use as crosslinker within the PMB system resulting in **N4-A4** and correspondingly PMB-Flex (Supplementary Fig. 5, Fig. 5a). PMB-Flex showed a shell thickness of ~ 230 nm (Supplementary Fig. 36) and were characterized by optical microscopy where no coalescence or rupture was observed for two months (Fig. 5b).

Sonication of PMB-Flex with US at 20 kHz resulted in an immediate development of fluorescence (Supplementary Fig. 37), which led us to investigate the fluorescence intensities of sonicated samples in dependence of different applied sound intensities (Supplementary Figs. 37 and 38). This yielded activated mechanophore fractions (Fig. 5c), with the released fraction increasing alongside the US intensity. With 20 kHz US at 12 W cm$^{-2}$ for 15 min, ~ 15% payload could be released comparing favorably to the original bulk polymer investigation of Boydston and coworkers[54]. Cross-validation by $^1$H NMR underlined this result (Fig. 5d). The application of 0.68 MHz US also successfully initiated the flex-activation release at $I \geq 75.5$ W cm$^{-2}$ (Supplementary Fig. 39, Fig. 5e).

To scrutinize these results, we synthesized the monofunctional mechanophore **N4-B4** as a control sample that would not be able to

undergo flex-activation, albeit being covalently copolymerized into the PMB shell (Supplementary Fig. 40). No fluorescence was observed under sonication conditions identical to above excluding thermal or other interfering effects. Moreover, water-soluble, mechanophore-centered linear polymers (LPs) with $M_n$ = 140 kDa were prepared to exclude the strong inertial cavitation of water as a source for the successful flex-activation. Also in this case sonication showed no fluorescence (Supplementary fig. 41) although the $M_n$ decreased to ~ 100 kDa due to unproductive bond scission. Likewise, no notable release could be discerned upon the sonication of solid microgel particles with diameters of ~ 25 μm containing **N4-A4** clarifying that contingent dangling, solvated chain ends do not contribute to the observed PMB performance (Supplementary figs. 42 and 43, Fig. 5f).

We then measured force-distance curves by atomic force microscopy (AFM) equipped with an FMV-A tip. PMBs and control microgels revealed considerable differences in shape deformation when loading the same 5 nN force with average deformations of microgels and PMBs being 2.3 nm and 60.5 nm, respectively (Fig. 5g). PMBs showed an average stiffness of ~ 300 pN nm$^{-1}$, while the stiffness of microgels was one order of magnitude larger with ~ 4900 pN nm$^{-1}$ (Fig. 5h and Supplementary fig. 44). This relatively larger deformability of the PMB shells likely contributed significantly to the oscillation of PMB volume induced by US.

An explanation of the flex-activation mechanism through combined experimental and computational studies was previously attempted by Boydston and coworkers and is the basis of the currently predominant interpretation of their observed results as caused by bond angle distortion[54]. To revisit the mechanistic interpretation of this activation mode, we first attempted to model the influence of the mechanical stress in PMBs by a pulling force, utilizing the approach of the "Force Modified Potential Energy Surface" (FMPES) according to Martínez and coworkers (Fig. 5i, j)[62]. In agreement with previously published literature, it became obvious that the simulation of a pulling force onto the system does not lead to a reduction of the activation energy of the flex-activation[63,64]. Even at high forces of 4.0 nN, only a reduction in activation energy $V_A$ of about 5.5 kcal mol$^{-1}$ was observed, reflecting the mechanochemical inactivity of linear chains in elongational flow fields[58]. Since the US-induced deformation of PMBs arguably leads to the generation of a pressure load on the polymer shell[50], a model based on the idea of the "Generalized Force Modified Potential Energy Surface"[65] (G-FMPES) method was developed and applied to simulate uniaxial pressure onto the molecular system adding an external potential $V_{ext}$ to the ab initio potential summing over all $N$ atoms:

$$V_{ext} = \frac{k}{2} \sum_{i=1}^{N} \cdot z_i^2 \qquad (1)$$

In Eq. 1, $z_i$ is the distance of the $i$-th atom to the plane created by the $x$- and the $y$-axes and $k$ is an external parameter describing the strength of the uniaxial pressure acting on the system. Thus, every atom of the system was pushed towards the $xy$-plane by using a harmonic potential (see Supplementary fig. 45 for illustration). The corresponding external force $\mathbf{F}_{ext}$ was then obtained by calculating

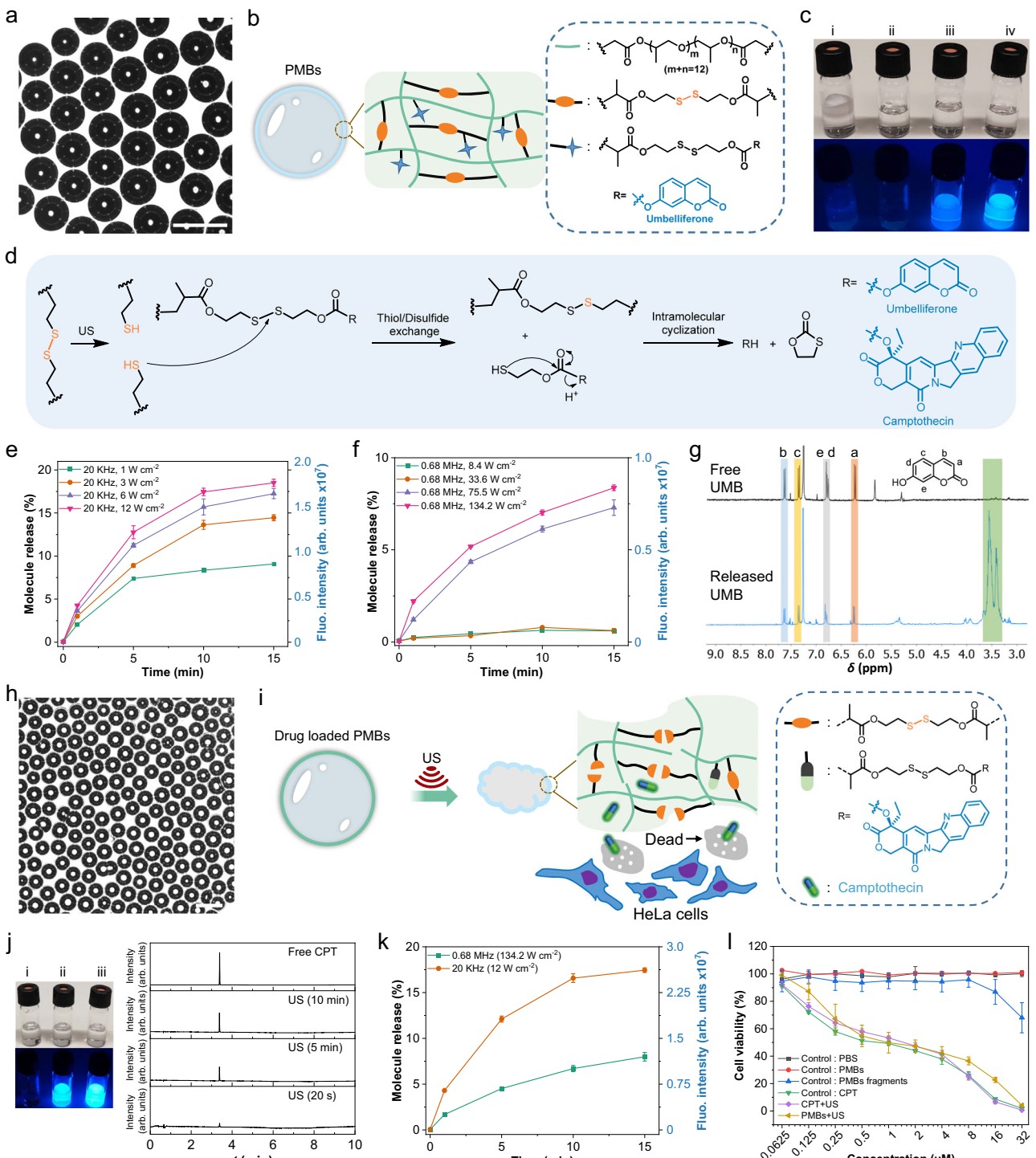

**Fig. 3 | The activation of disulfide mechanophores and small molecule release via thiol/disulfide exchange from PMBs. a** Optical micrograph, scale bar: 50 μm. The experiment was repeated three times, and similar images were obtained. **b** Structural composition of PMB-UMB shell. **c** Photographs of samples under daylight and UV light ($\lambda_{exc}$ = 365 nm), (i) PMB-UMB without sonication. (ii) DMSO/H$_2$O blank. (iii) The filtered solution of sonicated PMB-UMB (0.68 MHz, 15 min). (iv) The filtered solution of sonicated PMBs-UMB (20 kHz, 15 min). **d** Scheme of mechanophore activation and fluorophore release. **e, f** UMB release under 20 kHz and 0.68 MHz irradiations. Data are presented as mean values +/− the standard deviation. $N$ = 3 independent sonications. **g** $^1$H NMR spectra (400 MHz) of free UMB and released UMB. **h** Morphological characterization by optical microscopy, scale bar: 50 μm. The

experiment was repeated three times, and similar images were obtained. **i** Structural composition of PMB-CPT and illustration of CPT release. **j** Photographs of samples under daylight and UV light ($\lambda_{exc}$ = 365 nm), (i) DMSO/H$_2$O blank. (ii) The filtered solution of sonicated PMB-CPT (0.68 MHz, 15 min). (iii) The filtered solution of sonicated PMB-CPT (20 kHz, 15 min). UPLC elugrams of free CPT and released CPT (20 kHz). **k** CPT release from PMBs under 20 kHz and 0.68 MHz irradiation. Data are presented as mean values +/− the standard deviation. $N$ = 3 independent sonications. **l** MTS proliferation assay of negative controls PBS, PMBs and mechanophore-free PMB fragments, positive controls CPT and CPT + US, and ex situ sonicated (0.68 MHz, 15 min) PMB-CPT mixed with HeLa cells. Data are presented as mean values +/− the standard deviation. $N$ = 3 independent sonications. (Supplementary Table 3).

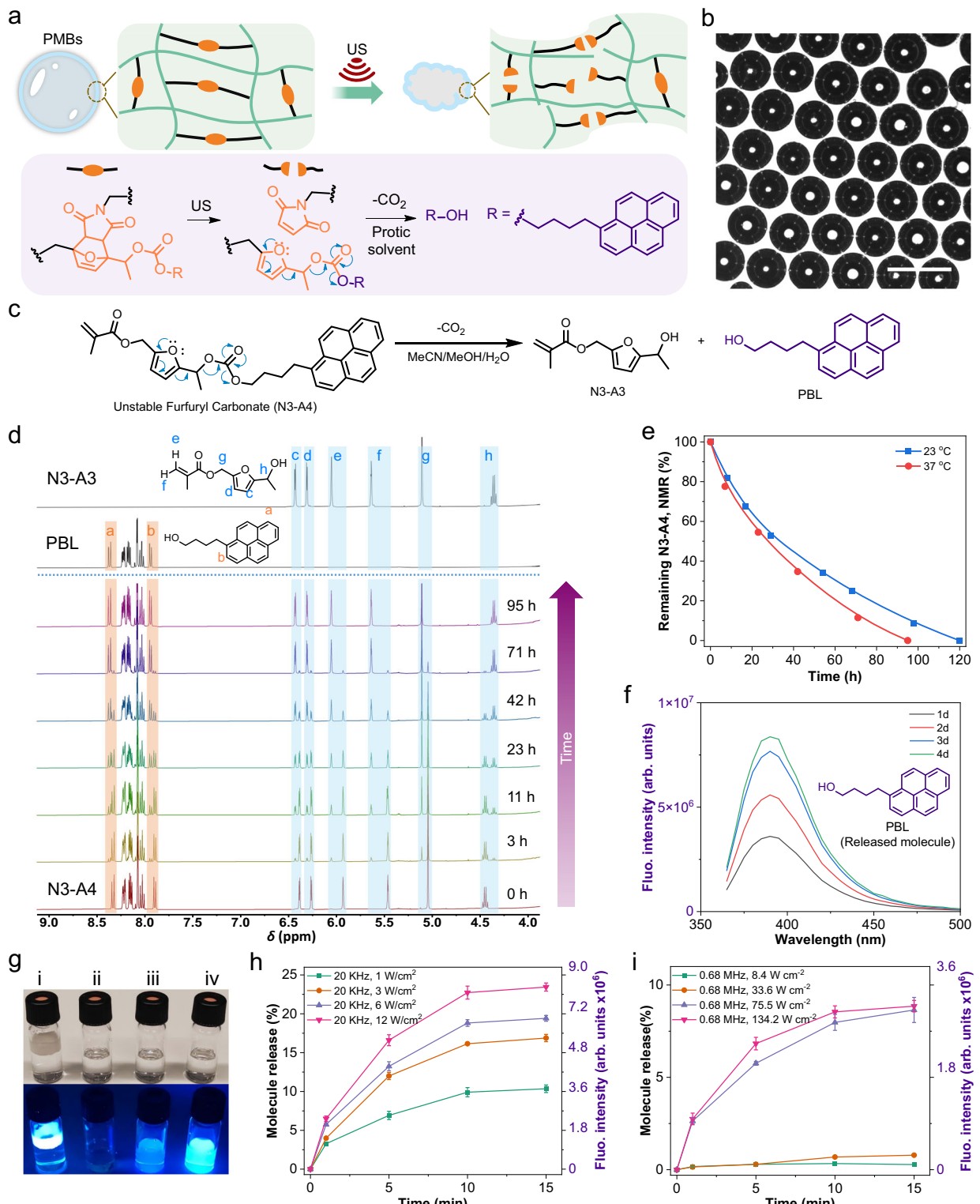

**Fig. 4 | US-triggered activation of 2-furylcarbinol maleimide Diels-Alder adducts and fluorogenic cargo release. a** Scheme of mechanophore activation in the shell of PMB-PBL. **b** Optical micrograph of PMB-PBL; scale bar: 50 μm. The experiment was repeated three times, and similar images were obtained. **c** The decomposition mechanism of **N3-A4** in polar protic solvent. **d** ¹H NMR (400 MHz) characterization of the conversion of **N3-A4** to **N3-A3** and PBL (solvent: MeCN-d₃:MeOH:H₂O, 3:1:0.5; 5 mg **N3-A4** in 0.5 mL solvent). **e** The time-dependent conversion of furfuryl carbonate **N3-A4** by ¹H NMR spectroscopy. **f** Fluorescence

spectra of sonicated PMB-PBL in MeCN:MeOH:H₂O (20 kHz, 12 W cm⁻²). **g** Photographs of samples under daylight and UV-light (λ_exc = 365 nm), (i) PMB-PBL solutions without sonication. (ii) MeCN/MeOH/H₂O blank. (iii) The filtered solution of sonicated PMB-PBL (0.68 MHz). (iv) The filtered solution of sonicated PMB-PBL (20 kHz). **h, i** The release of fluorogenic cargo under 20 kHz and 0.68 MHz irradiation. Data are presented as mean values +/− the standard deviation. N = 3 independent sonications.

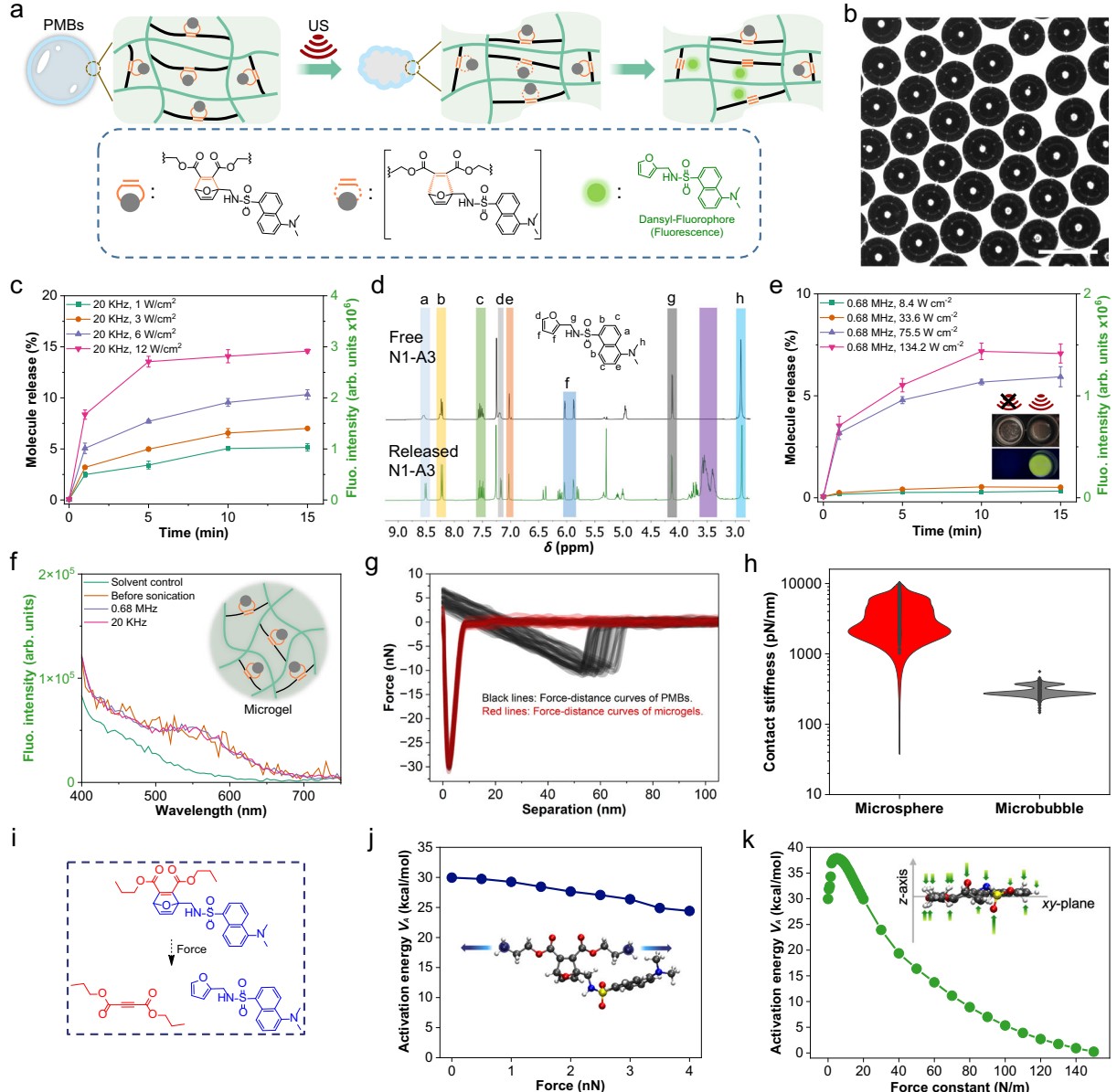

**Fig. 5 | US-triggered flex-activation of mechanophores in PMBs. a** Schematic illustration. **b** Optical micrograph of PMB-Flex, scale bar: 50 μm. The experiment was repeated three times, and similar images were obtained. **c** Fluorophore (**N1-A3**) release from PMB-Flex under 20 kHz US irradiation. Data are presented as mean values +/− the standard deviation. $N = 3$ independent sonications. **d** $^{1}$H NMR (400 MHz) spectra of free **N1-A3** and released **N1-A3**. **e** Fluorophore release from PMB-Flex under 0.68 MHz US irradiation. Inset: photographs of PMBs before (left) and after (right) HIFU treatment under daylight (top) and 365 nm UV (bottom)

illumination in MeCN/H$_2$O. Data are presented as mean values +/− the standard deviation. $N = 3$ independent sonications. **f** Fluorescence spectra of microgel suspension after 5 min US irradiation. **g** Force-distance curves of PMB-Flex and microgels obtained by AFM, scale bar: 50 μm. **h** Stiffnesses of PMB-Flex and microgels obtained from panel **g**. **i** Molecular structure that was used in computational modelling to investigate the flex activation. **j** Activation energy $V_A$ in dependence of the applied pulling force. **k** Activation energy $V_A$ in dependence of the force constant of the applied uniaxial pressure.

the negative derivative of the external potential $V_{ext}$ with respect to the spatial coordinates. This modification of the potential energy leads to flattened molecular structures, as can be seen in Supplementary fig. 46. In this study, the ab initio potential was calculated using density functional theory, see Methods. The optimized atomic coordinates of the electronic structure calculations are shown in Supplementary Data 1. Our computations indicate that a uniaxial pressure can reduce the activation energy $V_A$ (Fig. 5k), suggesting that other types of external forces rather than pulling forces may be responsible for the activation of the mechanophore. Using a force constant of 50 N m$^{-1}$, the activation energy $V_A$ is reduced to 16 kcal mol$^{-1}$, which would render a sufficiently small barrier for the reaction to occur.

In combination, our experimental and computational results on the one hand consolidate the findings of previous studies[54,58], and on the other hand arguably suggest that the action of US on PMBs does not exclusively result in stretching and fracture of the PMBs, but additionally compresses the polymer network architecture to a significant degree resulting in permanent damage and failure. This is underlined by the recent findings of Kiessling and coworkers who established that poly(n-butyl cyanoacrylate)- (PBCA)-shelled PMBs' US response is compression-dominated[66]. This would be in stark contrast to conventional US-based polymer mechanochemistry that relies on the overstretching of polymer segments in cavitation-induced flow fields[67]. Our findings on linear chains and microgels suggest that the uncoiling and overstretching of solvated chain segments is insufficient

to drive the flex-activation mechanism. Contrarily, the elastic cross-linked network structure, expected volume oscillation and implosion of PMBs, and the computationally demonstrated reduction of the activation barrier upon uniaxial compression indicate that compressive rather than extensional forces might affect the polymer system upon ultrasonication. The absence of activation of the monofunctional mechanophore is not in disagreement with this hypothesis since the accessible conformation space of a bifunctionally copolymerized mechanophore is considerably lower compared to its monofunctional counterpart inhibiting possible rotation of the molecule out of the direction of compression to avoid the energetically penalized bond angle distortion.

## Discussion

Overall, introducing gas into PMBs can enhance their collapse and generate compression forces, due to the pressure amplification effect of the gas core. When a gas-filled bubble is subjected to ultrasonic irradiation, the volume decreases rapidly during the compression (high-pressure) phase of the sound wave, which can quickly cause a sharp increase in pressure. The internal gas pressure rises dramatically and acts on the shell. When the pressure is high enough, the implosion happens. Therefore, this pressure generated by a gas core could enhance the PMBs' collapse compared to a PMB with a liquid or network core.

In addition, when the internal pressure gets very high, the implosion emits a shockwave into the surrounding fluid. The shockwave may act on the nearby polymeric bubbles. For the nearby bubbles that do not reach the implosion critical point, the shockwave and inertial cavitation may cause the nearby bubbles to explode.

The lower mechanophore activation yield under high frequency (MHz) irradiation is a challenge for PMBs' biomedical applications, such as drug delivery. When the frequency is above 1.5 MHz, the mechanophore activation yield is relatively low. Notably, 0.68 MHz irradiation with a sound intensity of 75.5 W cm$^{-2}$ led to 10% mechanophore activation, and the mechanical index (MI) is 1.8 (the FDA-approved safety margin MI is 1.9). Therefore, 0.68 MHz irradiation is a potential frequency for future biomedical applications of PMBs.

One suitable potential application for 20-30 μm bubbles is improving the sonothrombolysis of obstructive blood clots in large vessels or specific organs. The PMBs could be injected and used for directional blasting to loosen fibrin clots and promote thrombolytic medication diffusion, leading to faster and more precise clot treatment. For directional blasting, mechanophore activation is not needed; only bubble implosion. In this case, the 1.5 MHz irradiation (134.2 W cm$^{-2}$) could be used with a lower mechanical index (1.6). In addition, polyvinyl alcohol (PVA) and glycerol were used as surfactants to prepare PMBs. Both PVA and glycerol exhibit good biocompatibility. This could be beneficial for further medical translation of these PMBs.

We have utilized microfluidics to fabricate monodisperse microbubbles with a UV-curable polymer network shell, achieving uniform size and stability. The size of these PMBs can be precisely tuned by adjusting channel dimensions and flow parameters, providing a highly versatile platform. By introducing a wide variety of mechanophores into these PMBs, we underlined their versatility to incorporate diverse cargoes and mechanochemical activation pathways in the context of polymer mechanochemistry. Specifically, we showed that the PMB platform activates diverse mechanophore classes, including covalent disulfide scission, rDA reactions of furan-maleimide adducts, and the flex activation of furan-acetylene adducts. Through molecular computation, we proposed that the elastic polymer network shell, in conjunction with the oscillation of the PMB volume, likely exerts a compressive force on the polymer structure, contrasting with the elongational forces typically observed in solvated polymer chains during ultrasonication. This distinctive mechanism bridges the gap between solution-phase and bulk polymer network mechanochemistry.

We thus demonstrated that the PMB system enables multiple molecular pathways for the release of small-molecule cargo, with its efficient US response extending to high-frequency ranges up to the MHz scale. These findings highlight the PMB platform's potential for targeted drug delivery applications. Importantly, our system enables the integration of various payloads into the shell by covalent bonding, effectively mitigating cargo leakage – a common issue in other microbubble systems[68,69]. However, challenges such as optimizing PMB size for diagnostic and therapeutic applications remain to be addressed.

In summary, this work establishes a conceptual framework for integrating mechanochemical principles with PMB technology, paving the way for applications in both polymer chemistry and biomedical fields.

## Methods

### Preparation of polymeric microbubbles (G/O/W emulsion) by microfluidic engineering

High-purity N$_2$ was used as the gas phase to prepare the G/O/W double emulsion products-polymeric microbubbles[70]. N$_2$ was blown into the microfluidic device under the regulation of a flow rate controller (R100-02B, AirCom Pneumatic GmbH) with a digital pressure valve (MKA-08, AirCom Pneumatic GmbH). Then, the oil phase solution loaded into a glass syringe (Gastight Model 1001, Hamilton) was injected into the microfluidic device by a syringe pump (PHD ULTRA, Harvard Apparatus). The continuous aqueous phase was pumped into the microfluidic device through a dispensing pressure vessel (XX6700P01, Millipore Corporation). Once the flow rate of each phase reached the desired value, the double emulsion production in the microfluidic device commenced automatically. The products (PMBs) were collected in a glass vial and washed 3 times using DI H$_2$O before sonication treatment. G/O/W emulsion: Gas: N$_2$; Oil phase: PPGDA, PPG, TPO-L, toluene, and the variable mechanophore comonomer; H$_2$O phase: PVA (4.5%, w/w) and glycerol (40%, v/v). The setting of microfluidic station: gas (530 mbar), oil phase (90 nL min$^{-1}$), H$_2$O phase (2.3 bar). A high-power UV LED (LZ1-10UV0R-0000, Osram) was clamped above the outlet tube to initiate the polymerization of the oil-phase shell of the PMBs.

### Preparation of microgels (O/W emulsion)

Microgels were prepared by O/W single emulsion in the microfluidic device. The oil phase solution was injected into the microfluidic device by a syringe pump (PHD ULTRA, Harvard Apparatus). Then, the aqueous phase solution was pumped into the microfluidic device through a dispensing pressure vessel (XX6700P01, Millipore Corporation). Once the flow rate of each phase reached the desired value, the single emulsion production in the microfluidic device commenced automatically. The microgels were collected in a glass vial and washed 3 times using DI H$_2$O before sonication treatment. O/W emulsion: Oil phase: PPGDA, PPG, TPO-L, and the variable mechanophore comonomer; H$_2$O phase: PVA (3%, w/w). The setting of microfluidic station: Oil phase (150 nL min$^{-1}$), H$_2$O phase (0.27 bar). A high-power UV LED (LZ1-10UV0R-0000, Osram) was clamped above the outlet tube to initiate the polymerization of the oil-phase.

### Sonication experiments

Before sonication, the number concentration of PMBs (1.9 10$^7$ mL$^{-1}$) and microgels (6.4 10$^6$ mL$^{-1}$) were regulated using an EVE Automatic cell counter (NanoEntek). Briefly, PMBs (100 μL) or microgels (30 μL) were mixed with 1 mL solvent, then transferred to a 1.5 mL Eppendorf tube or 24 well plate (lumox multiwell 24, Sarstedt). Next, the samples were sonicated for 1, 5, 10, or 15 min. $f$ = 20 kHz: Experiments were

carried out using a Qsonica Q500 ultrasonic system with four 3.2 mm probes (A12628PRB20), pulsed sonication (2 s on, 1 s off). The sound intensity $I$ of applied US was calculated by $I = P \cdot A^{-1}$, where $P$ is the output power obtained via the spent energy over the sonication on-time $t$ ($P = E \cdot t^{-1}$) and $A$ is the surface area of the transducer probe ($A = \pi r^2$). HIFU: Experiments were performed with a home-built HIFU setup. The core devices include waveform generator (33511B, Keysight Technologies), RF amplifier (AG1021, T&C Power Conversion), and transducers (0.68 MHz, 1.52 MHz, and 2.6 MHz, Precision Acoustics). A 0.5 mm needle hydrophone (Precision Acoustics) was used for locating the transducer's focal point. Custom-made motorized 3D-positioning system for controlling the well plate submerged in water was employed. The focal point of the transducer was located at the axis of the well, 3 mm above its bottom. RF-voltage was applied to the transducer to obtain the desired focal pressure according to the transducer's factory calibration. Following the approach of Precision Acoustics Ltd., we estimated the intensity to be the total acoustic power divided by the transvers focal area (whose boundary is determined at -6 dB level). For sinusoidal waves (in a linear approximation) acoustic pressure (p) is related to the intensity ($I$), $I = p^2/\rho c$, where $\rho$ is fluid density and c is speed of sound. Pulsed sonication (2 s on, 1 s off) was used.

**MTS proliferation assays.** Cell viability was studied by culturing HeLa cells with different samples in a basal medium containing DMEM (supplemented with 10% fetal bovine serum and 1% antibiotics/antimycotics) at 37 °C. The measurement of cell viability was implemented by using a commercial kit, which contain tetrazolium compound 3-(4,5-dimethylthiazol-2-yl)-5-(3-carboxymethoxyphenyl)-2-(4-sulfophenyl)-2H-tetrazolium (inner salt, MTS reagent) and a chemical electron acceptor dye (phenazine ethosulfate; PES) (Promega, Germany). Briefly, approximately 5000 cells in 100 µL of medium were seeded into 96-well plates. After overnight incubation, the culture medium in 96-well plates was removed and exchanged with fresh medium (100 µL) containing different concentrated testing samples. Control cultures were treated with DMSO alone. The final concentration of DMSO in the medium was controlled by less than 0.5%. After 48 h incubation, cell culture media were removed, the cells were rinsed with 100 µL PBS buffer, and then 20 µL MTS reagent with 100 µL fresh medium was added to the cells. The resulting solution was mixed thoroughly, and the absorbance was measured using a microplate spectrophotometer at 490 nm (Synergy™ HT microplate reader, Bio-Tek Instruments). MTS signals represented the survival and proliferation determination. The mixture of MTS reagent with cell culture medium was served as negative control. All the sample cultures were performed at least in triplicates. HeLa cells (CCL-2 ™) used in this study were obtained from DWI – Leibniz Institute for Interactive Materials. HeLa cells (CCL-2 ™) were originally purchased from ATCC: The Global Bioresource Center.

**Atomic force microscopy (AFM) measurements**
AFM measurements of the PMBs and microgels were performed in the dry state, at the silicon wafer-air interface, using a Dimension Icon with a closed loop (Veeco Instruments, and Software: Nanoscope 9.4, Bruker). The data were recorded in force volume mode (FV) using FMV-A tips with a resonance frequency of 75 kHz, and a nominal tip radius of 7 nm (Bruker). An OPUS AFM probe of the aluminium coated 240AC-NA cantilever was calibrated in air using the thermal resonance method built into the AFM software (spring k = 0.9 N/m). Contact stiffness is measured by indenting the apex of the microbubble or microsphere at 1µm/s over an area of (10×10) µm and analyzed by extracting the slope of the force-distance plots, i.e. the force increment as the tip indents the surface to a depth of 40 nm with a maximum force load of 5 nN.

**Computations**
Computations were performed on the level of density functional theory (DFT) employing the B3LYP functional[71–73] using D3-dispersion correction[74] and the 6-31 G* basis set[75–77]. TeraChem[78,79] was utilized for computing the electronic structure. Stationary points under the influence of external forces were obtained using DL-FIND[80], which is interfaced to TeraChem via a python interface. This python interface allows to incorporate a module simulating the effects of a pulling force as well as the effects of a uniaxial acting force. The convergence criterion for the gradient was set to $1.0 \times 10^{-4}$ atomic units. To confirm whether the geometry found is a minimum on the (force-modified) potential energy surface, the eigenvalues of the Hessian matrices were considered, as for minima the Hessian matrix is positive definite. For the optimization of transition state structures, the dimer method was used[81,82]. The found structures were confirmed to be first order saddle points on the (force-modified) potential energy surface by the Hessian matrices as they show exactly one negative eigenvalue in case of a transition state. For calculations utilizing the FMPES approach, the external force was increased stepwise in increments of 0.5 nN up to 4.0 nN. For simulating a uniaxial pressure onto a system, the force constant $k$ of the external harmonic potential was increased in general in increments of 10 N m$^{-1}$. Below 20 N m$^{-1}$, smaller increments were chosen to achieve better resolution on the effects of lower uniaxial pressure onto a molecular system.

**Software used in the study**
Data collection/data analysis: ChemDraw 19.0, OriginPro 2024b, MestReNova V 14.2.3-29241, PSS Win GPC UniChrom V 8.32, Nanoscope Analysis V 1.9, TeraChem V 1.9, FlyCapture 2.11.3.164, LAS X V 3.7.0.20979, SoftMax Pro V 7.0.3, Microsoft Excel and PowerPoint 2016, OriginPro 2024b, MestReNova V 14.2.3-29241, Chem3D 19.0.1.28.

**Reporting summary**
Further information on research design is available in the Nature Portfolio Reporting Summary linked to this article.

# Data availability
All data supporting the findings of this study are documented within manuscript and Supplementary Information, and the Source Data are openly available in Zenodo at https://doi.org/10.5281/zenodo.15622722. Data is available from the corresponding authors on request.

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

## Acknowledgements

Funding is acknowledged from the German Research Foundation for 503981124 (R.G.) and 464121872 (A.H.), the Werner Siemens Foundation (grant 4D direct in-wound printing for functional tissue repair (Trigger-INK)) (A.H.) and the EU through an ERC Advanced Grant (SONO-PHARMAGEN, No. 101142296) (A.H.). Moreover, this work has been funded as part of the Leibniz ScienceCampus: ACTISONO, supported by the Leibniz Association grant no. W89/2023 (A.H.). Moreover, Funding from the Leibniz Association through the Leibniz Research Alliance 'Advanced Materials Safety' is acknowledged (A.H.). R.L. and J.M. acknowledge the support of computational infrastructure provided by the Centre for Information and Media Technology at Heinrich Heine University Düsseldorf. R.L. is grateful for the support by the Jürgen Manchot Foundation. J.M. is grateful for a materials cost allowance from the Fonds der Chemischen Industrie. We thank Rostislav Vinokur, the constructor of the HIFU system.

## Author contributions

J.F., M.X., R.G. and A.H. conceived the project. J. F. developed the experimental methods. J.F. performed the experiments and analyzed the results. R.L. and J.M. performed and analyzed the computations for flex-activation. J.F. and M.X. conducted the data analyses with support of R.G. A.M. conducted AFM experiments. K.Z. analyzed the AFM data and gave suggestions for drawing figures. A.H. acquired funding and supervised the project. J.F. drafted the manuscript. M.X., R.G. and A.H. revised the manuscript. All authors participated in the writing process.

## Funding

## Competing interests

The authors declare no competing interests.
