## [Transparent Peer Review file · Nature Communications]

Polymer microbubbles as universal platform to accelerate polymer mechanochemistry

Corresponding Author: Professor Andreas Herrmann

Version 0:

Reviewer comments:

Reviewer #1

(Remarks to the Author)

The authors demonstrate PMBs as a platform for accelerated and enhanced mechanochemical activation of various mechanophores using both 20 kHz and MHz ultrasound. Particularly, this platform allows release of small molecules upon mechanochemical activation induced by biocompatible HIFU (MHz frequency), which is a step forward toward the practical application of polymer mechanochemistry in biomedical fields. And remarkably, this is the first report that flex-activation can be achieved by ultrasonication, though in crosslinked polymers rather than in solvated linear polymers.

The research looks solid and complete, and the conclusion are supported by the results. I recommend its publication in Nat. Commun. after minor revisions. A few specific comments are listed below:

1. The shell thickness is given only in the case of flex-activated mechanophores. According to a previous report by the same group (Adv Mater 2023, 35, 2305130), the PMB shell thickness affects mechanochemical activation, therefore it is important to report the thickness values in the other cases.
2. The caption in Fig. 1 does not match with the corresponding panels (e.g., a and b), and the caption for panel c is missing.

Reviewer #2

(Remarks to the Author)

The authors present a versatile and efficient PMB-based platform for activating three types of mechanophores using ultrasound of varying frequency and intensity. This work serves as an important complement and extension to their previous study (Adv. Mater. 35, 2305130 (2023)). Their investigation establishes a universal and optimized platform for activating both scissile and flex-activated mechanophores, which together represent the majority of mechanophore systems. Notably, they propose a pressure-induced activation mechanism in contrast to the conventional tensile-stretching mechanism. Furthermore, molecular dynamics simulations on flex-activated mechanophores were conducted to elucidate the underlying activation mechanism.

Overall, I strongly support the publication of this work in Nature Communications. However, I have several suggestions for further improving the manuscript before final acceptance:

1. Cavitation Mechanism and Bubble Behavior:

The activation of mechanophores within PMBs is closely linked to cavitation events. It would be beneficial for the authors to include a discussion on the cavitation mechanism, particularly how introducing N₂ gas into PMBs enhances bubble collapse and generates compression forces. Moreover, to distinguish whether the PMBs undergo an implosion or explosion mechanism, the frequent use of the term "burst" in the manuscript (which suggests an explosion mechanism) should be clarified. Could the authors provide supporting evidence, such as optical micrographs of PMBs before and after sonication, to validate their claim?

2. Incorporation of Non-Scissile Mechanophores:

Mechanophores generally fall into three categories based on their activation mechanism: scissile, non-scissile, and flex-activated. The authors focus on scissile and flex-activated mechanophores but do not investigate non-scissile ones. Since the mechanical properties of PMBs typically change upon scission-based mechanophore activation, incorporating non-scissile mechanophores could potentially maintain the overall mechanical integrity of PMBs while leading to a higher mechanophore activation percentage. This effect has been widely observed in non-scissile linear multi-mechanophore systems. I wonder if the authors could explore a similar phenomenon in PMBs. A suitable candidate for this study is the

alkoxy-based gem-dichlorocyclopropane (J. Am. Chem. Soc. 2020, 142, 99–103), which can be synthesized in just two steps. Since its polymer backbone remains intact after activation, the released acid could be quantified using an acid-sensitive dye to determine activation efficiency.

3. Standardization of Ultrasound Irradiation Descriptors:

In the sentence:

“At 20 kHz and an intensity of $I = 12 \text{ W cm}^{-2}$, sonication for 5 min led to bursting of the PMBs, and the fragments sank to the bottom of the tube. Under HIFU irradiation, PMB explosion commenced at a focal pressure of 1300 kPa for 0.68 MHz, 1800 kPa for 1.52 MHz irradiation, and 6000 kPa for 2.6 MHz irradiation.”

The manuscript switches between intensity (W cm^{-2}) and pressure (kPa) to describe ultrasound conditions. For consistency and clarity, could the authors use a uniform descriptor throughout?

4. Clarification of Figures:

o Adding the ultrasound frequency to Figures 2h–k would improve clarity.

o In Figure 5g, the meaning of the black and red lines should be explicitly defined, as not all readers may be familiar with their significance.

5. Impact of Ultrasound Frequency on Mechanophore Activation and Biocompatibility:

The manuscript reports that increasing ultrasound frequency results in a lower mechanophore activation percentage at the same focal pressure. How does this trend affect potential applications in biocompatible mechanochemistry? A discussion on the implications for biomedical and therapeutic applications would be valuable, particularly regarding both activation efficiency and the biocompatibility of materials used in this platform.

6. Definition of N₂-CL:

The term “N₂-CL” should be clearly labeled and properly defined in the manuscript for clarity.

7. Influence of PMB Size and Shell Thickness:

Could the authors summarize how PMB size, particularly diameter and shell thickness, affects the observed outcomes under ultrasound irradiation based on their findings?

I look forward to reading a revised version of this manuscript.

Reviewer #3

(Remarks to the Author)

This manuscript explores the feasibility of incorporating four different mechanophores into PMBs to enable ultrasound-induced mechanochemical activation. Building upon their prior work (Adv. Mater. 35, 2305130, 2023), the authors demonstrate that PMBs can effectively activate mechanophores under both low-frequency (20 kHz) and high-frequency (MHz) ultrasound by converting acoustic energy into compressive mechanical forces. This study highlights the versatility of PMBs as acousto-mechanical transducers and introduces flex-activation under ultrasound, which may have implications for biomedical applications. Overall, the manuscript presents a promising strategy for ultrasound-responsive mechanochemistry and generally fits the scope of Nat. Commun. I suggest the authors address the following points prior to publication.

1. The thickness of the PMB shell is not mentioned for all the PMBs. It remains unclear whether all PMBs share the same shell thickness, whether it was optimized, and whether variations in thickness impact the activation efficiency of embedded mechanophores.

2. The activation mechanism also needs further discussion. It is not clearly established whether mechanophore activation results from PMB deformation, rupture, or both. Post-sonication characterization is needed to determine whether intact PMBs persist after sonication and whether fragments can continue to activate mechanophores. This would help distinguish the role of dynamic versus static mechanical forces in the observed activation.

3. The relatively low activation yield (typically <20%) raises questions regarding the mechanistic limitations. It would be helpful for the authors to discuss potential reasons for this incomplete conversion. Additionally, the reduced activation efficiency at higher ultrasound frequencies is noted but not well-explained. Are there differences in fragment morphology, size, or structure that correlate with activation outcomes under different conditions?

4. The stability of PMBs after synthesis is also not addressed. Clarifying whether the PMBs must be used immediately or can be stored, and whether aggregation occurs over time, would be important for reproducibility. Furthermore, the manuscript would benefit from details regarding the optimized concentration of PMBs used in activation experiments and the rationale for selecting high-purity N₂ as the internal gas phase.

5. The computational work of the flex-activated mechanophore using G-FMPES does not clearly align with the entire manuscript narrative. The authors suggest that compression is a favorable activation mode for flex-mechanophore; however, the computationally based conclusion is somewhat speculative. A more informative approach may involve probing the fluidity and viscosity of the PMB core to understand how deformation translates into mechanical force on the shell. It is also important to clarify whether the core in the microgel control experiments is fully gelled/solidified or retains a liquid character. Exploring cores with varying viscosities could yield more conclusive evidence for the proposed activation mechanism.

Version 1:

Reviewer comments:

Reviewer #1

(Remarks to the Author)

Since all my concerns have been addressed in the revised manuscript, I recommend its publication as it is.

Reviewer #2

(Remarks to the Author)

Since the authors have demonstrated their capability in synthesizing non-scissile mechanophores, and all remaining comments have been adequately addressed, I support the acceptance of the paper.

Reviewer #3

(Remarks to the Author)

The authors have provided a thorough and thoughtful revision of the manuscript, including additional experiments and an expanded discussion that effectively addresses all of my previous concerns. I am satisfied with the revisions. I recommend it for publication without further revision.

Dear Reviewers,

We are thankful for being given the opportunity to submit a revised version of our work and for the discussion with Reviewers, which we believe has improved the quality of our manuscript.

With my best wishes,

Andreas Herrmann

The following are point-by-point responses to the comments:

Reviewer #1

Comment 1.1: The authors demonstrate PMBs as a platform for accelerated and enhanced mechanochemical activation of various mechanophores using both 20 kHz and MHz ultrasound. Particularly, this platform allows release of small molecules upon mechanochemical activation induced by biocompatible HIFU (MHz frequency), which is a step forward toward the practical application of polymer mechanochemistry in biomedical fields. And remarkably, this is the first report that flex-activation can be achieved by ultrasonication, though in crosslinked polymers rather than in solvated linear polymers.

The research looks solid and complete, and the conclusion are supported by the results. I recommend its publication in Nat. Commun. after minor revisions. A few specific comments are listed below:

Reply 1.1: We thank the Reviewer for the very positive and kind words. We are thankful for the Reviewer's appreciation of this work.

Comment 1.2: The shell thickness is given only in the case of flex-activated mechanophores. According to a previous report by the same group (Adv Mater 2023, 35, 2305130), the PMB shell thickness affects mechanochemical activation, therefore it is important to report the thickness values in the other cases.

Reply 1.2: Thank you for your recognition and interest in our previous work. Indeed, the shell thickness affects mechanochemical activation. In this paper, we use the same microfluidic parameters (Flow rate: gas (530 mbar), oil phase (90 nL min⁻¹), H₂O phase (2.3 bar)) and the same batch of chips to prepare four kinds of microbubbles. The thickness of the other three kinds of bubbles was measured by SEM. SEM images showed that there was no significant difference in shell thickness. The shell thickness of the four kinds of bubbles is very similar, around 230 nm. The shell thickness information of the other three kinds of bubbles was added in the supporting information (Figure S7, Figure S23, and Figure S28).

Comment 1.3: The caption in Fig. 1 does not match with the corresponding panels (e.g., a and b), and the caption for panel c is missing.

Reply 1.3: We thank the Reviewer for their observant comment. The caption in Figure 1 was revised accordingly (page 3).

Reviewer #2

Comment 2.1: The authors present a versatile and efficient PMB-based platform for activating three types of mechanophores using ultrasound of varying frequency and intensity. This work serves as an important complement and extension to their previous study (Adv. Mater. 35, 2305130 (2023)). Their investigation establishes a universal and optimized platform for activating both scissile and flex-activated mechanophores, which together represent the majority of mechanophore systems. Notably,

they propose a pressure-induced activation mechanism in contrast to the conventional tensile-stretching mechanism. Furthermore, molecular dynamics simulations on flex-activated mechanophores were conducted to elucidate the underlying activation mechanism.

Overall, I strongly support the publication of this work in Nature Communications. However, I have several suggestions for further improving the manuscript before final acceptance:

Reply 2.1: We thank the Reviewer for the in-depth review and the very positive feedback. We have attempted to answer the Reviewer's comments or questions in the following replies.

Comment 2.2: Cavitation Mechanism and Bubble Behavior:

The activation of mechanophores within PMBs is closely linked to cavitation events. It would be beneficial for the authors to include a discussion on the cavitation mechanism, particularly how introducing N₂ gas into PMBs enhances bubble collapse and generates compression forces. Moreover, to distinguish whether the PMBs undergo an implosion or explosion mechanism, the frequent use of the term "burst" in the manuscript (which suggests an explosion mechanism) should be clarified. Could the authors provide supporting evidence, such as optical micrographs of PMBs before and after sonication, to validate their claim?

Reply 2.2: We appreciate the Reviewer's detailed comments on the cavitation mechanism. Introducing gas into PMBs can enhance their collapse and generate compression forces, due to the pressure amplification effect of the gas core.

The gas core has a pressure amplification effect in bubble collapse. When a gas-filled bubble is subjected to ultrasonic irradiation, especially during the compression (high-pressure) phase of the sound wave, it collapses very rapidly. During this implosion, the internal gas pressure rises dramatically—far beyond the pressure applied by the ultrasound. Due to the fact that the gas inside obeys the ideal gas law, the volume decreases rapidly, which causes a sharp increase in pressure. This could be well explained by the PV^γ constant formula.

$PV^\gamma = \text{constant}$ (PV^γ constant formula)

P = pressure inside bubble

V = volume of the bubble

γ = adiabatic index

From the PV^γ constant formula, we know that as V shrinks, P increases. As the PMB has a decreased volume of the gas core, a huge pressure from the gas core will act on the shell. When the pressure is high enough, the burst (it's implosion) happens during the compression phase. Therefore, this pressure generated by a gas core could enhance PMBs collapse compared to polymeric architectures with a liquid or network core (no pressure or low pressure generation).

In addition, when the internal pressure gets extremely high (maybe reaching several to hundreds of atmospheres), the implosion causes a shockwave to be emitted into the surrounding fluid. The shockwave could also act on nearby polymeric bubbles. For the nearby bubbles that do not reach the implosion critical point, the shockwave and inertial cavitation may cause the nearby bubbles' 'explosion'.

This is underlined by the recent findings from Kiessling and coworkers, who established that the PBCA (poly(*n*-butyl cyanoacrylate)) shelled PMBs' ultrasound response is compression-dominated (ACS Appl. Bio Mater. 2025, 8, 2, 1240–1250. <https://pubs.acs.org/doi/10.1021/acsbm.4c01551>). Optical micrographs of PBCA-shelled microbubbles revealed either a compression-dominated or compression-

only response at peak negative acoustic pressures higher than 165 kPa. This finding supports our speculation that the action of ultrasound on PMBs does not exclusively result in stretching and bursting but additionally compresses the polymer network architecture to a significant degree.

Overall, we believe that the existing literatures and our own experiments support that polymer-shelled microbubbles can burst due to expansion-contraction movements in the ultrasound field. The word “burst” on page 12 of the manuscript was replaced with ‘implosion’. Moreover, we termed the effect “fracture” where the implication is only the generation of PMB fragments and their secondary degradation, but not whether an implosion or explosion occurred.

We added three paragraphs (pages 4 and 13) to the Results and Discussion section of the manuscript to discuss the mechanophore activation mechanism.

Comment 2.3: Incorporation of Non-Scissile Mechanophores:

Mechanophores generally fall into three categories based on their activation mechanism: scissile, non-scissile, and flex-activated. The authors focus on scissile and flex-activated mechanophores but do not investigate non-scissile ones. Since the mechanical properties of PMBs typically change upon scission-based mechanophore activation, incorporating non-scissile mechanophores could potentially maintain the overall mechanical integrity of PMBs while leading to a higher mechanophore activation percentage. This effect has been widely observed in non-scissile linear multi-mechanophore systems. I wonder if the authors could explore a similar phenomenon in PMBs. A suitable candidate for this study is the alkoxy-based gem-dichlorocyclopropane (J. Am. Chem. Soc. 2020, 142, 99–103), which can be synthesized in just two steps. Since its polymer backbone remains intact after activation, the released acid could be quantified using an acid-sensitive dye to determine activation efficiency.

Reply 2.3: We thank the Reviewer for this constructive suggestion. We just developed this polymeric microbubble system over the past two years. Our group's research on polymer mechanochemistry is mainly focused on sonopharmacology. So we first tried the mechanophores that could be used for molecule or drug release. That's why disulfide, 3,2-furylcarbinol Diels-Alder adduct mechanophore, and flex-activated mechanophores were used in this paper. Our focus is currently on the study of ultrasound-induced drug release. In addition, ozone must be used for the suggested synthesis. Ozone is a harmful gas and not easy to obtain (an ozonizer is usually needed).

In the future, we plan to enhance the PMB system to other mechanophores. The suggestion to use non-scissile mechanophores, such as gDHCs, is therefore a great inspiration for us to maintain PMB integrity while activating many mechanophores.

Comment 2.4: Standardization of Ultrasound Irradiation Descriptors:

In the sentence:

“At 20 kHz and an intensity of $I = 12 \text{ W cm}^{-2}$, sonication for 5 min led to bursting of the PMBs, and the fragments sank to the bottom of the tube. Under HIFU irradiation, PMB explosion commenced at a focal pressure of 1300 kPa for 0.68 MHz, 1800 kPa for 1.52 MHz irradiation, and 6000 kPa for 2.6 MHz irradiation.”

The manuscript switches between intensity (W cm^{-2}) and pressure (kPa) to describe ultrasound conditions. For consistency and clarity, could the authors use a uniform descriptor throughout?

Reply 2.4: We thank the Reviewer for pointing out this issue. For a 20 KHz sonicator, W cm^{-2} is usually used to describe the sound intensity. For focused ultrasound, the focal pressure kPa is generally used to describe the sound intensity. We provided the conversion relationship in the supporting information (Table S2). Now, for consistency and clarity, we uniformed the unit. W cm^{-2} is used to describe the sound intensity in the whole manuscript and supporting information.

Comment 2.5: Clarification of Figures:

- o Adding the ultrasound frequency to Figures 2h–k would improve clarity.
- o In Figure 5g, the meaning of the black and red lines should be explicitly defined, as not all readers may be familiar with their significance.

Reply 2.5: We appreciate the Reviewer's suggestion. To better understand and read this article, the ultrasound frequencies were added to Figure 2h–k. The meaning of the black and red lines was defined in Figure 5 g.

Comment 2.6: Impact of Ultrasound Frequency on Mechanophore Activation and Biocompatibility:

The manuscript reports that increasing ultrasound frequency results in a lower mechanophore activation percentage at the same focal pressure. How does this trend affect potential applications in biocompatible mechanochemistry? A discussion on the implications for biomedical and therapeutic applications would be valuable, particularly regarding both activation efficiency and the biocompatibility of materials used in this platform.

Reply 2.6: We thank the Reviewer for the attention to the potential biomedical applications of PMB. When the frequency is above 1.5 MHz, the mechanophore activation yield is relatively low. Actually, the lower mechanophore activation yield under high (MHz) frequency irradiation is a challenge for PMB's applications, such as for drug delivery. Based on our knowledge, the irradiation frequency above 1 MHz is not suitable for the drug release of our polymeric bubbles. Notably, 0.68 MHz irradiation with a sound intensity of 1500 KPa (or 75.5 W cm⁻²) led to 10 % mechanophore activation, and the mechanical index (MI) is 1.8 (the FDA-approved safety margin MI is 1.9). Therefore, 0.68 MHz irradiation is a potential frequency for future biomedical applications of PMBs.

One potential application for 20-30 micron-sized bubbles is improving the sonothrombolysis of obstructive blood clots in large vessels or specific organs. The PMB could be injected and used for 'directional blasting' to loosen fibrin clots and promote thrombolytic medication diffusion, leading to faster and more precise clot treatment. In addition, for 'directional blasting', mechanophore activation is not needed; only the bubble burst is necessary. In this case, the 1.5 MHz irradiation (2000 KPa or 134.2 W cm⁻²) could be used, due to irradiation with a lower mechanical index (1.6). In addition, PVA and glycerol were used as surfactants to prepare PMBs. Both PVA and glycerol exhibit good biocompatibility (Adv. Funct. Mater. 2023, 33, 2212952; European Polymer Journal. 2022, 175, 111377). Therefore, the PMBs should have good biocompatibility and potential biomedical applications. Our research is currently in the early stages of biomedical applications. We will perform more attempts in the future.

We added two paragraphs (page 14) to the Results and Discussion section of the manuscript to discuss the potential biomedical applications.

Comment 2.7: Definition of N₂-CL:

The term "N₂-CL" should be clearly labeled and properly defined in the manuscript for clarity.

Reply 2.7: We express our gratitude for this very kind reminder. "N₂-CL" is a spelling error; it's 'N1-CL'. We have corrected this mistake in the supporting information (page S45).

Comment 2.8: Influence of PMB Size and Shell Thickness:

Could the authors summarize how PMB size, particularly diameter and shell thickness, affects the observed outcomes under ultrasound irradiation based on their findings?

I look forward to reading a revised version of this manuscript.

Reply 2.8: We are thankful for this in-depth comment from the reviewer. Based on our findings, both diameter and shell thickness affected the mechanophores activation in polymeric microbubbles.

Diameter affects the mechanophore activation by changing the eigenfrequency of the polymeric bubble. We found that bubbles are easily destroyed when the frequency of ultrasound irradiation is close to the eigenfrequency of the polymeric bubbles. In our previous study (Adv. Mater. 2023, 35, 2305130), the computational investigations using near-resonance excitation supported our findings, suggesting that the US-induced deformation of the PMB shells was largest when the frequency of the ultrasound source approached the eigenfrequency of the PMB. For the PMB with a size around 25 μm , its eigenfrequency is around 21 KHz; our result shows that 20 KHz (12 W cm^{-2}) irradiation leads to a higher mechanophore activation. Using high frequency focused ultrasound at megahertz (far off the eigenfrequency of PMB) with the same sound intensity (12 W cm^{-2}) didn't lead to the mechanophore activation, which underlined this interpretation.

Shell thickness also affected the mechanophores activation in polymeric microbubbles. Theoretical calculations in our previous study (Adv. Mater. 2023, 35, 2305130) show that an increased PMB deformation occurs with the thinner shell thickness upon ultrasonication. The PMBs with a thicker shell are mechanically robust and can resist mechanical deformation. Larger deformation could lead to more mechanophore activation during the deformation and lead to fracture processes.

In addition, after the destruction of PMBs, the fragments from the thicker shell can't be torn into smaller pieces by cavitation. This also led to the lower mechanophore activation yield of thicker shelled PMBs.

Overall, we found that thin shells and the PMB size-dependent eigenfrequency close to the irradiation frequency maximized the mechanophore activation.

Reviewer #3

Comment 3.1: This manuscript explores the feasibility of incorporating four different mechanophores into PMBs to enable ultrasound-induced mechanochemical activation. Building upon their prior work (Adv. Mater. 35, 2305130, 2023), the authors demonstrate that PMBs can effectively activate mechanophores under both low-frequency (20 kHz) and high-frequency (MHz) ultrasound by converting acoustic energy into compressive mechanical forces. This study highlights the versatility of PMBs as acousto-mechanical transducers and introduces flex-activation under ultrasound, which may have implications for biomedical applications. Overall, the manuscript presents a promising strategy for ultrasound-responsive mechanochemistry and generally fits the scope of Nat. Commun. I suggest the authors address the following points prior to publication.

Reply 3.1: We also thank this reviewer for the positive feedback. We have attempted to address the Reviewer's concern in the following replies.

Comment 3.2: The thickness of the PMB shell is not mentioned for all the PMBs. It remains unclear whether all PMBs share the same shell thickness, whether it was optimized, and whether variations in thickness impact the activation efficiency of embedded mechanophores.

Reply 3.2: Thank you for pointing out this issue. In the current manuscript, we have only given the shell thickness of the flex-activated PMBs (see Figure S31). The manuscript didn't show the shell thicknesses of the other microbubbles.

In this paper, we use the same microfluidic parameters (Flow rate: gas (530 mbar), oil phase (90 nL min^{-1}), H_2O phase (2.3 bar)) and the same batch of microfluidic chips to prepare the four kinds of microbubbles. The thickness of the other three kinds of bubbles was measured by SEM. SEM images showed that there was no significant difference in shell thickness. The shell thickness of the four kinds

of bubbles is very similar, around 230 nm. The shell thickness information of the other three kinds of bubbles was added in the supporting information (Figure S7, Figure S23, and Figure S28).

Indeed, the variations in shell thickness affect the mechanophore activation. Theoretical calculation in our previous study (Adv. Mater. 2023, 35, 2305130) shows that an increased PMB deformation occurs with the thinner shell thickness upon ultrasonication. Large deformation leads to high mechanophore activation. The PMB with a thicker shell has a sturdy construction that can resist mechanical deformation. Furthermore, after the burst of PMB, the fragments from the thicker shell are difficult to be torn into smaller pieces by cavitation. This also led to the lower mechanophore activation yield for PMB with thicker shell thickness.

Comment 3.3: The activation mechanism also needs further discussion. It is not clearly established whether mechanophore activation results from PMB deformation, rupture, or both. Post-sonication characterization is needed to determine whether intact PMBs persist after sonication and whether fragments can continue to activate mechanophores. This would help distinguish the role of dynamic versus static mechanical forces in the observed activation.

Reply 3.3: We are thankful for this in-depth review. Based on our results, mechanophore activation occurred in all three processes: PMB deformation, rupture, and subsequent sonication of PMB fragments.

To investigate if the PMB deformation process led to mechanophore activation, we performed a verification experiment. 5 min sonication with a very low sound intensity (20 KHz, 0.2 W cm^{-2}) was performed. Only PMB deformation occurred upon this sonication process, and PMBs kept the intact structure with this low sound intensity. Finally, we found around 3% disulfide mechanophore activation within 5 min sonication. This result was added to the supporting information, Figure S13.

PMB rupture certainly led to the activation of mechanophore. During the rupture process, the mechanophores at the cracks between fragments can be activated due to the pull force between the adjacent pieces. Furthermore, the gas core has a pressure amplification effect leading to bubble collapse. When a gas-filled bubble is subjected to ultrasonic irradiation, especially during the compression (high-pressure) phase of the sound wave, it collapses very rapidly. During this implosion, the internal gas pressure rises dramatically—far beyond the pressure applied by the ultrasound. Because gas inside obeys the ideal gas law, the volume decreases rapidly during the compression phase and quickly causes a sharp increase in pressure. The pressure from the gas core will act on the shell, which could also lead to mechanophore activation (please see **Reply 2.2** for pressure amplification effect in bubble collapse).

After the PMB rupture, the collision and deformation of fragments during the rupture process can also cause the mechanophore activation. From figure 2h, we know that around 8% mechanophore was activated within 1 min sonication, around 23% mechanophore was activated within 15 min sonication. In addition, we found that the PMBs didn't survive more than 1 min sonication (20 KHz , 12 W cm^{-2}), see figure 2e. This means the subsequent 14 min sonication still led to an increase in mechanophore activation. This was also indirectly confirmed by the fact that the fragments gradually become smaller during the 15 min sonication (see figure 2e, the three images above).

We characterized the PMBs after sonication; no intact PMBs were present after 1 min of sonication (20 KHz , 12 W cm^{-2}); see Figure 2e.

We added three paragraphs (pages 4 and 13) to the Results and Discussion section of the manuscript to discuss the mechanophore activation mechanism.

Comment 3.4: The relatively low activation yield (typically <20%) raises questions regarding the mechanistic limitations. It would be helpful for the authors to discuss potential reasons for this incomplete conversion. Additionally, the reduced activation efficiency at higher ultrasound frequencies is noted but not well-explained. Are there differences in fragment morphology, size, or structure that correlate with activation outcomes under different conditions?

Reply 3.4: We again thank the Reviewer for this in-depth review. Compared to linear polymers, which have a high mechanochemical conversion (usually above 50% to quantitative), PMBs have a lower activation yield, which may be due to their network structure. For linear polymers, every polymer molecule is dissolved in the solution and experiences the shear forces generated by cavitation. The weakest molecular link (mechanophore) is positioned where force is statistically most likely to break the polymer chain, either with or without mechanophore.

Conversely, polymer networks are known to undergo mostly random bond scission. Force is not always transmitted to the formally weakest molecular link (the mechanophore), but network defects, such as dangling chain ends, entanglements or other inhomogeneities, lead to partially random bond scission on the most overstretched chain segments. Compared to bulk network mechanophore activation, PMBs perform extraordinarily well, since in such experiments (compare works from Creton and coworkers) only single digit percentage bond scission fractions could be measured.

In addition, the collision of PMB fragments in the solution will happen under sonication. After PMB rupture, the mechanophores more centrally located in PMB fragments may receive protection from the surface layer. During sonication, the inner layer avoids collision, and the collision of PMB fragments only happens on the surface layer. This could also reduce the activation yield. PMBs with a thin shell have higher activation, which seems to confirm this.

For the reduced activation efficiency at higher ultrasound frequencies, there are two reasons. One reason is the used irradiation frequency of 20 KHz close to the eigenfrequency of the PMBs. In our previous study (Adv. Mater. 2023, 35, 2305130), the computational investigations suggested that the US-induced deformation of the PMB shells was largest when the frequency of the ultrasound source approached the eigenfrequency of the PMB. For the PMBs with a size around 25 μm , its eigenfrequency is around 21 KHz; so, 20 KHz irradiation leads to a larger deformation of the PMB shells, which reflects a higher mechanophore activation. Using high frequency focused ultrasound at megahertz (MHz far off the eigenfrequency of PMB), the deformation of the PMB shells was smaller, and the smaller deformation led to the low activation efficiency. Another reason is that the irradiation with high frequency (MHz) generates low cavitation. After the fragmentation of PMBs, the fragments from the shell are torn into smaller pieces to a lesser extent by low cavitation. This also led to the lower mechanophore activation yield.

The fragments' morphology of PMB after three different frequency (0.68, 1.52, 2.6 MHz) irradiations was characterized, see figure 2e (the three images below). We could see that the PMBs maintain a more complete bubble structure after 2.6 MHz irradiation under the same sound intensity (134.2 W cm^{-2} , 15 min). The mechanophore activation results show that the activation is lowest under 2.6 MHz irradiation. The fragments' morphology after 20 kHz sonication (see Figure 2e above) also confirmed this: The activation gradually increases with the fragments becoming smaller. Overall, we found that the higher mechanophore activation is accompanied by smaller-sized fragments. Bigger-sized fragments represent lower mechanophore activation.

Comment 3.5: The stability of PMBs after synthesis is also not addressed. Clarifying whether the PMBs must be used immediately or can be stored, and whether aggregation occurs over time, would be important for reproducibility. Furthermore, the manuscript would benefit from details regarding the

optimized concentration of PMBs used in activation experiments and the rationale for selecting high-purity N₂ as the internal gas phase.

Reply 3.5: We thank the reviewer for this very detailed review. The PMBs have excellent stability. PMBs don't need to be used immediately. They can be stored at room temperature for at least 2 months, and no aggregation occurs over time.

We didn't optimize the concentration of PMBs used in the mechanophore activation experiments. In the previous paper (*Adv. Mater.* 2023, 35, 2305130), 10 μl PMBs (in 1 mL solvent) was used for the experiments. In this manuscript, 100 μl PMBs (in 1 mL solvent) was used for the activation experiments. We found no significant difference in mechanophore activation efficiency between two samples with different concentration of PMBs; the disulfide mechanophore activation was around 22% (20 kHz). Here, we use 100 μl PMBs in the experiments to collect a higher fluorescence intensity value, which may improve the sensitivity and accuracy of fluorescence intensity detection.

The PMBs with the N₂ core have excellent stability. We also tried to use air as gas to prepare bubbles. However, the bubbles' stability is poor, and their structure is prone to denting; this may be due to the fact that the air contains multiple ingredients (such as oxygen and carbon dioxide), and some gas is dissolved in the oil phase. High-purity N₂ was selected as the internal gas phase due to its low solubility in the oil phase.

Comment 3.6: The computational work of the flex-activated mechanophore using G-FMPES does not clearly align with the entire manuscript narrative. The authors suggest that compression is a favorable activation mode for flex-mechanophore; however, the computationally based conclusion is somewhat speculative. A more informative approach may involve probing the fluidity and viscosity of the PMB core to understand how deformation translates into mechanical force on the shell. It is also important to clarify whether the core in the microgel control experiments is fully gelled/solidified or retains a liquid character. Exploring cores with varying viscosities could yield more conclusive evidence for the proposed activation mechanism.

Reply 3.6: We thank the reviewer for the concern about the hypothetical activation mode of the flex-mechanophore.

Other groups and this study verified that the flex-mechanophore incorporated in the linear polymer cannot be activated by shear forces generated by ultrasound. This result confirmed that the traditional shear force (two directions along the polymer chain) is ineffective for 'flex-mechanophores'. To the best of our knowledge, by now, all the reported 'flex-mechanophore' activations are by compression on the elastomeric materials or by cryo-milling for centimeter-size network polymers (*J. Am. Chem. Soc.* 2013, 135, 8189–8192; *J. Am. Chem. Soc.* 2014, 136, 1276–1279; *Polymer*, 2018, 152, 4-8; *Angew. Chem. Int. Ed.* 2023, 62, e202308662; *ACS Macro Lett.* 2025, 14, 14–19; *Polym. Chem.*, 2022, 13, 3986). Therefore, we have reason to assume that the deformation and destruction of PMBs results not only in stretching but also in compressive force. Then, the generated compressive force acts on the polymer-network shell leading to the 'flex-mechanophore' activation.

Furthermore, the gas core has a pressure amplification effect leading to bubble collapse (please see **Reply 2.2** for pressure amplification effect).

In combination, our experimental and computational results arguably suggest that the action of ultrasound on PMBs does not exclusively result in stretching and bursting but additionally compresses the polymer network architecture to a significant degree. This is underlined by the recent findings from Kiessling and coworkers, who established that the PBCA (poly(*n*-butyl cyanoacrylate)) PMBs' ultrasound response is compression-dominated (*ACS Appl. Bio Mater.* 2025, 8, 2, 1240–1250. <https://pubs.acs.org/doi/10.1021/acsbm.4c01551>). Optical micrographs of PBCA-shelled

microbubbles revealed either a compression-dominated or compression-only response at peak negative acoustic pressures higher than 165 kPa. This finding supports our speculation that the action of ultrasound on PMBs does not exclusively result in stretching and bursting but additionally compresses the polymer network architecture.

Based on our results, we agree with the reviewer's claim that the fluidity and viscosity of the core can affect the mechanophore activation in the shell. In our previously published paper (*Adv. Mater.* 2023, 35, 2305130), we prepared the microgels and microcapsules as control samples. The core of the three samples (bubbles, microgels and capsules) has different fluidity and viscosity. However, we found that the mechanophore activation only happens in the bubbles with the gas core. No mechanophore activation was detected in microcapsules (PMB with a water core). Therefore, we speculate that a liquid core or a polymer network core will not benefit mechanophore activation in the shell. Only bubbles (with a gas core) respond sensitively to ultrasonic irradiation.

In addition, the core in the microgel is fully gelled. The generated microgels from the microfluidic channel pass through an outlet tube, and UV light is set to surround the outlet tube. All the microgels that pass through the outlet tube have undergone UV irradiation for at least 10 min. We performed a test in which the 1 ml bulk solution (for microgel preparation) was placed in an Eppendorf tube for gelation. The solution gelled after 3 min of the same UV irradiation. Therefore, 10 min UV irradiation is enough for the gelation of micro-sized microgels.

Overall, we again express our gratitude for these very fruitful comments.